# AI-Based Noise-Reduction Filter for Whole-Body Planar Bone Scintigraphy Reliably Improves Low-Count Images

**DOI:** 10.3390/diagnostics14232686

**Published:** 2024-11-28

**Authors:** Csaba Csikos, Sándor Barna, Ákos Kovács, Péter Czina, Ádám Budai, Melinda Szoliková, Iván Gábor Nagy, Borbála Husztik, Gábor Kiszler, Ildikó Garai

**Affiliations:** 1Division of Nuclear Medicine and Translational Imaging, Department of Medical Imaging, Faculty of Medicine, University of Debrecen, H-4032 Debrecen, Hungary; csikosc@med.unideb.hu (C.C.); barna.sandor@scanomed.hu (S.B.); czinapeter@gmail.com (P.C.); nagy.ivan@med.unideb.hu (I.G.N.); 2Gyula Petrányi Doctoral School of Clinical Immunology and Allergology, Faculty of Medicine, University of Debrecen, H-4032 Debrecen, Hungary; 3Scanomed Ltd., H-4032 Debrecen, Hungary; budai.adam@scanomed.hu; 4Mediso Ltd., H-1037 Budapest, Hungary; akos.kovacs@mediso.com (Á.K.); melinda.szolikova@mediso.com (M.S.); borbala.husztik@mediso.com (B.H.); gabor.kiszler@mediso.com (G.K.)

**Keywords:** bone scan, nuclear medicine, imaging, noise reduction filter, artificial intelligence

## Abstract

**Background/Objectives**: Artificial intelligence (AI) is a promising tool for the enhancement of physician workflow and serves to further improve the efficiency of their diagnostic evaluations. This study aimed to assess the performance of an AI-based bone scan noise-reduction filter on noisy, low-count images in a routine clinical environment. **Methods**: The performance of the AI bone-scan filter (BS-AI filter) in question was retrospectively evaluated on 47 different patients’ ^99m^Tc-MDP bone scintigraphy image pairs (anterior- and posterior-view images), which were obtained in such a manner as to represent the diverse characteristics of the general patient population. The BS-AI filter was tested on artificially degraded noisy images—75, 50, and 25% of total counts—which were generated by binominal sampling. The AI-filtered and unfiltered images were concurrently appraised for image quality and contrast by three nuclear medicine physicians. It was also determined whether there was any difference between the lesions seen on the unfiltered and filtered images. For quantitative analysis, an automatic lesion detector (BS-AI annotator) was utilized as a segmentation algorithm. The total number of lesions and their locations as detected by the BS-AI annotator in the BS-AI-filtered low-count images was compared to the total-count filtered images. The total number of pixels labeled as lesions in the filtered low-count images in relation to the number of pixels in the total-count filtered images was also compared to ensure the filtering process did not change lesion sizes significantly. The comparison of pixel numbers was performed using the reduced-count filtered images that contained only those lesions that were detected in the total-count images. **Results**: Based on visual assessment, observers agreed that image contrast and quality were better in the BS-AI-filtered images, increasing their diagnostic confidence. Similarities in lesion numbers and sites detected by the BS-AI annotator compared to filtered total-count images were 89%, 83%, and 75% for images degraded to counts of 75%, 50%, and 25%, respectively. No significant difference was found in the number of annotated pixels between filtered images with different counts (*p* > 0.05). **Conclusions**: Our findings indicate that the BS-AI noise-reduction filter enhances image quality and contrast without loss of vital information. The implementation of this filter in routine diagnostic procedures reliably improves diagnostic confidence in low-count images and elicits a reduction in the administered dose or acquisition time by a minimum of 50% relative to the original dose or acquisition time.

## 1. Introduction

Whole-body bone scintigraphy is a routine diagnostic tool in the staging of different types of malignant diseases, like prostate cancer, lung cancer, and breast cancer. Furthermore, it is a widely used imaging method in various benign conditions (e.g., arthritis or infection) [1,2].

Its relative affordability and easy applicability makes bone scan an appealing modality for clinicians [3]. However, the sheer amount of cases, sometimes comprised of myriads of images to be assessed, can indeed be burdensome for nuclear medicine physicians [4], and the inherent noise in scintigraphy images can be detrimental to diagnostic accuracy. To address this issue, advanced noise-reduction techniques have been developed, among which artificial intelligence (AI)-based approaches have shown great promise.

Nowadays, AI and deep learning techniques have revolutionized medical imaging and image processing by enhancing image quality and supporting the diagnostic processes. Convolutional neural networks (CNNs) have been successfully applied in a variety of imaging modalities—including MRI, CT, and PET scans—to reduce noise and improve image resolution [5,6,7]. Kang et al. demonstrated the potential of a deep learning-based denoising algorithm in low-dose CT scans, which significantly improved image quality without compromising diagnostic performance [8]. Neural networks (NNs) learn the characteristic structures and patterns in a controlled training process [5]. Autoencoder is a family of NNs that was successfully used for medical image denoising; it samples the input image to a lower resolution in multiple stages, then restores the original image from this encoded representation and preserves the anatomical structures while eliminating the random noise component from the image [9].

Despite the much-welcomed features and advancements in machine learning technology, the employment of AI-based noise-reduction filters in bone scintigraphy is yet to be adopted on a more extensive basis. Reading bone scintigraphy images can present a challenge for physicians due to their high sensitivity and low specificity; the reason for which lies partly in the amount of noise present in the image [10]. Conventional noise-reduction techniques, such as Gaussian smoothing and median filtering, often cover up image details, potentially masking the clinically relevant features. Therefore, there is a need for robust AI-based solutions designed especially for the requirements of bone scintigraphy.

In this study, we evaluate the performance and the clinical utility of an AI-based noise-reduction filter that was trained to enhance the quality of the typically grainy whole-body bone scintigraphy images. This filter utilizing the advantages of neural networks may be a potential tool to effectively reduce noise while preserving important diagnostic information.

However, a performance evaluation of this filter is essential to ensure its reliability and effectiveness in clinical settings. The current study aimed to evaluate the performance of this AI-based noise-reduction filter using a comprehensive evaluation framework. This included assessment of its impact on lesion detectability, image contrast, and overall image quality compared to unfiltered images. By employing a rigorous evaluation approach, we sought to establish the clinical utility of this AI-based filter and its potential to improve diagnostic accuracy in whole-body bone scintigraphy.

## 2. Methods

### 2.1. Data Set

The training process of the neural network, which utilized data from 1215 patients, was described earlier in the article by Kovács et al. (2022) [11].

An independent image set was defined from archived, anonymized patient images for performance evaluation. The scans were performed using 500–600 MBq of Tc-99m methylene diphosphonate (99mTc-MDP) (Isotope Institute Ltd., Budapest, Hungary). The radiopharmaceutical was administered intravenously, followed by an accumulation time of 2–5 h. Imaging was carried out using the AnyScan DUO double head, or TRIO triple head SPECT/CT systems (Mediso Ltd., Budapest, Hungary). Images were acquired with a matrix size of 256 × 1024 and a scanning speed of 130 mm/min.

For assessment, 47 pairs of anterior- and posterior-view whole-body planar bone scans were selected. The image pairs were sampled in a uniform manner using an automatic DICOM metadata monitoring tool called Q-bot (version 1.7.2) [12]. One image was randomly drawn from each category, with the exclusion of those images utilized in the training process. The selection categories were defined based on the possible combinations of the following subcategories:Body mass index (BMI): underweight, normal, and obese patients;Age: <45 years, 45–65 years, and >65 years;Gender: female and male;Lesions: normal accumulation only, ≤3 lesions, and >3 lesions (multiple metastases).

Seven of the possible categories (54) were not included in the evaluation process because no such images could be retrieved from our image set.

### 2.2. Visual Assessment

Three nuclear medicine physicians independently compared the filtered images to the unfiltered (original) images and scored them based on four specific queries:Is there any lesion that is not visible in the original image but highlighted by the AI filter? (1: No, 2: Yes);Is there any lesion that is missed in the AI-filtered image? (1: No, 2: Yes);The contrast of the lesions in the filtered image: (1: Much Worse, 2: Poorer, 3: Equal, 4: Better, 5: Much Better);Overall assessment of the image quality of the AI-filtered image: (1: Much Worse, 2: Poorer, 3: Equal, 4: Better, 5: Much Better).

During the evaluation, we focused on the divergent and uncertain responses provided by the experts.

### 2.3. Automatic Assessment

A lesion detecting software component (BS-AI annotator) was applied for automatic, objective performance assessment purposes. The BS-AI annotator is based on the Mask R-CNN architecture [8], a deep learning model capable of delineating lesion boundaries at the pixel level by defining their bounding boxes and classifying them. Furthermore, it has the potential to be used for supporting medical diagnostic decision making, as demonstrated by recent results [5,9,10].

The BS-AI annotator consists of three main components: a preprocessor, a deep learning model, and a postprocessor (Figure 1). The algorithm requires anterior and posterior pairs of bone-scan DICOM images as input data. The preprocessor converts DICOM images to RGB format, then the deep learning model predicts the lesion masks and classifies them as one of three possible lesion types (likely benign, likely malignant, and artifact), after which the postprocessor refines the results by creating outline annotations and eliminating the overlapping masks. Overlapping masks with the same lesion types are automatically merged, while those with different lesions are forwarded to conduct a probability analysis. The only mask retained is the one with the highest probability score.

The BS-AI annotator processes anterior/posterior scan pairs, but predictions are made independently for each scan. Thus, predictions made on an anterior image do not use information from the posterior image and vice versa. The BS-AI annotator is deterministic, providing consistent results for the same input, making it suitable for our objective evaluation purpose.

The deep learning model of the BS-AI annotator was trained with 1664 pairs of whole-body bone scans. Scan specifications were identical to those described previously in Section 2.1.

The BS-AI annotator was utilized to objectively assess the abilities of the BS-AI filter to maintain structures. Lesion detection was performed in the filtered images. Furthermore, artificially degraded noisy images were generated using binomial sampling, resulting in images with counts reduced to 75%, 50%, and 25% of the total-count (100%) images, which were then filtered using the BS-AI filter tool. The lesion-detection process was executed in the degraded filtered images (Figure 2).

The annotator’s ability to detect lesions in the filtered images at different levels of degradation was compared to its lesion-detection ability in the filtered total-count images. Lesions identified in the total-count filtered images were defined as true positives (TP; the reference standard). Lesions annotated in the filtered degraded images but not found in the filtered total-count images were defined as false positives (FP), and lesions annotated in the filtered total-count images but not in the filtered degraded images were defined as false negatives (FN). Similarities to the total-count images, i.e., TP’s/(TP’s + FP’s + FN’s), were given in percentages.

Size differences of the annotated lesions between images degraded to various extents were analyzed at the pixel level. The number of annotated pixels was compared in images where no false-positive (FP) or false-negative (FN) lesions were identified. Similarities in the number of annotated pixels between the low-count and total-count filtered images are expressed in percentages. These percentages were then compared using one-way ANOVA, followed by a post hoc Tukey test.

## 3. Results

### 3.1. Results of the Visual Assessment

During the course of visual assessment, no vanishing or new lesions were found by the nuclear medicine physicians on the filtered 100%-, 75%-, and 50%-count images compared to their respective unfiltered image pairs. However, two physicians identified a disappearing lesion in an image degraded to a count of 25% after the filter was applied (Figure 3B). Similarly, two physicians detected one image that highlighted a new lesion after filtering, which was not visible in the unfiltered image (Figure 3A).

In most cases, the contrast of the lesions in the filtered images was perceived to be better or much better than in the unfiltered images (Figure 4A). Overall image quality was also found to be better or much better in the filtered images in the majority of cases (Figure 4B).

### 3.2. Results of the Automatic Segmentation

To avoid bias due to physicians recalling higher-count images of the same patient, the BS-AI filter’s capability of maintaining lesions on images of varying counts was assessed by using the deterministically working BS-AI annotator. Since three nuclear medicine physicians agreed that the filtered total-count images showed exactly the same lesions as the unfiltered total-count images, filtered total-count images were used as the reference standard. Assessment of similarities in lesion detection by the BS-AI annotator indicated that artifacts were detected most consistently compared to the filtered total (100%)-count (C100%) images as image quality was degraded to lower counts of 75% (C75%), 50% (C50%), and 25% (C25%). Similarities in artifact assessment between the degraded filtered images and the filtered total-count images were 98%, 93%, and 91% for C75%, C50%, and C25% images, respectively (Figure 5A). It is particularly important to note that the corresponding similarity values for likely benign lesions were 89%, 84%, and 76% and 88%, 80%, and 71% for likely malignant lesions, respectively (Figure 5B,C). The overall similarities to C100% images in lesion detection were 89%, 83%, and 75%, respectively (Figure 5D).

The number of pixels annotated in images where no FP or FN lesion was found in any of the three different count images was compared. Similarities to C100% images in the number of pixels marked as lesions (implies lesion sizes indirectly) are given in ratios. Images degraded to different extents did not show any significant difference in pixel numbers after filtering and annotating (Figure 6).

## 4. Discussion

AI-based filters offer tremendous advantages over traditionally used nuclear imaging filters. Unlike conventionally used low-pass and band-pass filters, NN-based image filters can produce noise-reduced images with high contrast. However, extra caution must be taken when such filters are developed. Previous research projects have proven that CNN-based image filters outperform conventional filters in diagnostic performance; therefore, in this article, we did not examine the their performance [13,14]. Despite their ability to generate high-contrast images, there is a risk that NN-based filters potentially eliminate true abnormal uptake, leading to false negatives, or create artificial lesions from noise, resulting in false positives [13,14,15]. Therefore, besides the assessment of nuclear medicine physician opinions, integrating objective methods into the performance evaluation process is essential.

Data from the literature show that whole-body bone SPECT has a better specificity than planar bone scintigraphy [16,17]; however, planar bone scan remains the gold standard diagnostic modality for screening metastatic bone lesions due to its affordable cost and wide availability [1,18]. Images from bone scintigraphy often contain Poisson noise that can make the interpretation process difficult and can worsen the diagnostic value of the examination [13]. These findings emphasize the need for planar bone scintigraphy and the need for image quality-enhancing solutions, like CNN-based image filters.

Denoising software using deep learning is available for several nuclear medicine examinations: bone scan [13,14,15], 18F-FDG PET [19,20,21,22], 68Ga-PSMA PET [23], 64Cu-DOTATATE PET [24], 68Ga-DOTATOC PET [25], and Tc-99m DMSA [26]. The strength of our evaluation strategy in contrast to, for instance, the evaluation of a noise-reduction filter by Minarik et al. [13] is that not only do subjective results (i.e., physician questionnaire) support the appropriate function of the BS-AI filter, but a deterministically working AI-based lesion-detecting software was also applied to objectively analyze the filter’s noise-reducing and contrast-enhancing capability. Murata et al. [14] used an AI-based lesion-detecting software called BONENAVI in order to evaluate whether false-positive or -negative findings were present after the filtering process. BONENAVI (mainly used in Japan) and EXINI (mainly used in Europe and North America) are both types of artificial neuronal network-utilizing image analysis software. Their main function is the application of the bone scan index (BSI), which is a useful tool to evaluate therapeutic response and prognosis [27].

This current study evaluated the AI-based noise-reduction filter developed by Kovács et al. [11] through both visual assessment and automatic software detection. Subjective assessment of the filtered images demonstrated that disappearing and newly created lesions are uncommon, and their presence is debatable. Only two of these cases were found amongst the 25%-count images by two physicians each. All the three physicians agreed that there was no vanishing or falsely appearing lesion after filtering the 50%-, 75%-, and 100%-count images. Physicians agreed that filtered images had better contrast and overall quality compared to the unfiltered images. As shown in Figure 3, during the assessment of contrast and overall quality of the filtered images, a great heterogeneity was seen between the individuals, which is due to the physicians’ subjective interpretation. This finding highlights the importance of an objective evaluation approach.

Objective assessment with a lesion-detection software revealed that even in the images degraded to 25% of the original count, a 75% similarity to the full-count images could be achieved in the annotations. A comparison with the pixel counts of the filtered total-count images revealed that there was no significant difference in the number of pixels detected between filtered images whose counts were reduced to varying extents. The lesion-detection software that we used in the evaluation is a deterministically working program, which means it finds the exact same lesions if an image input is used multiple times. This characteristic of the program makes it suitable for the detection of newly appearing and vanishing accumulation signals.

The evaluation of the BS-AI filter suggests that the underlying concept of this CNN deep learning model has the potential to serve as a reliable tool for training AI-based noise-reduction filters. The implementation of this algorithm for SPECT is also feasible. This, however, presupposes an adequate training process on an appropriate number of tomographic images.

## 5. Conclusions

The results of our study indicate that this BS-AI filter has the potential to be a valuable tool for physicians, aiding their workflow by allowing the administration of a considerably reduced dose of activity or the minimization of acquisition time. Our findings suggest that reducing the injected dose or acquisition time by at least half of the originally administered activity or acquisition time is a reasonable approach.

## Figures and Tables

**Figure 1 diagnostics-14-02686-f001:**
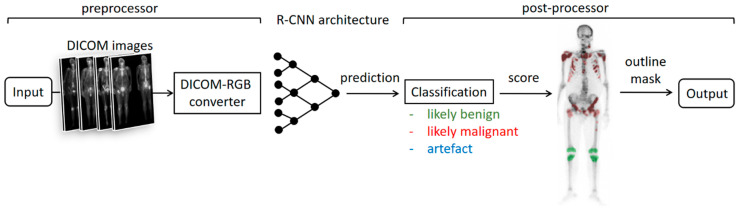
High-level architecture of the BS-AI annotator software component.

**Figure 2 diagnostics-14-02686-f002:**
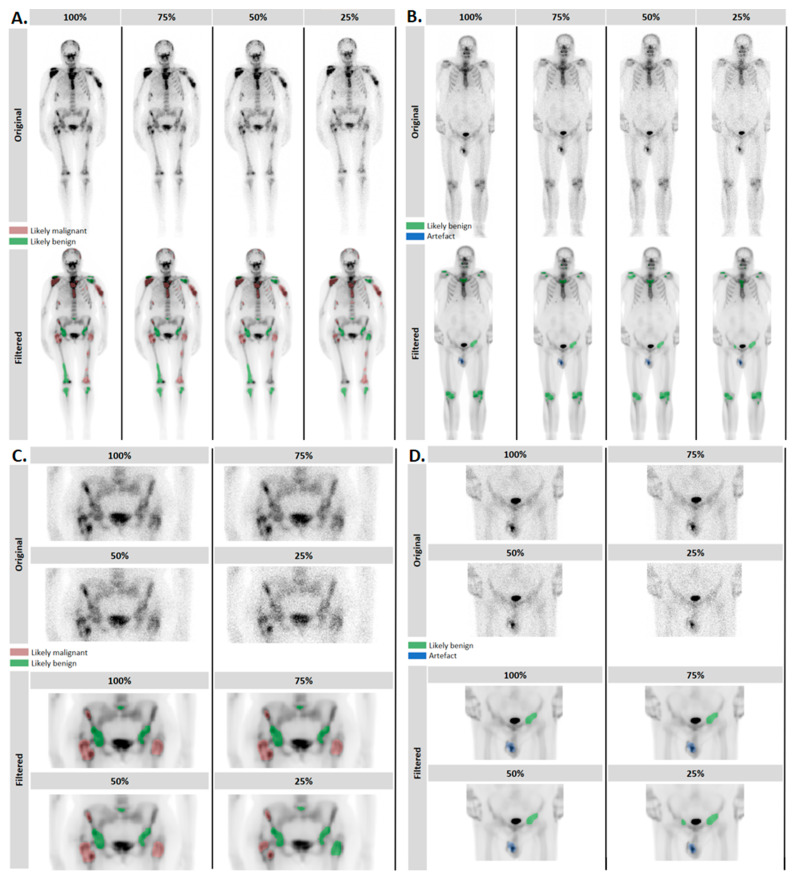
Original images with degraded image quality and filtered images of a normal BMI (**A**) and an obese patient (**B**). Lesions marked with different colors show the findings of the BS-AI annotator. Pelvic areas of the same patients are shown for detailed visualization (**C**,**D**).

**Figure 3 diagnostics-14-02686-f003:**
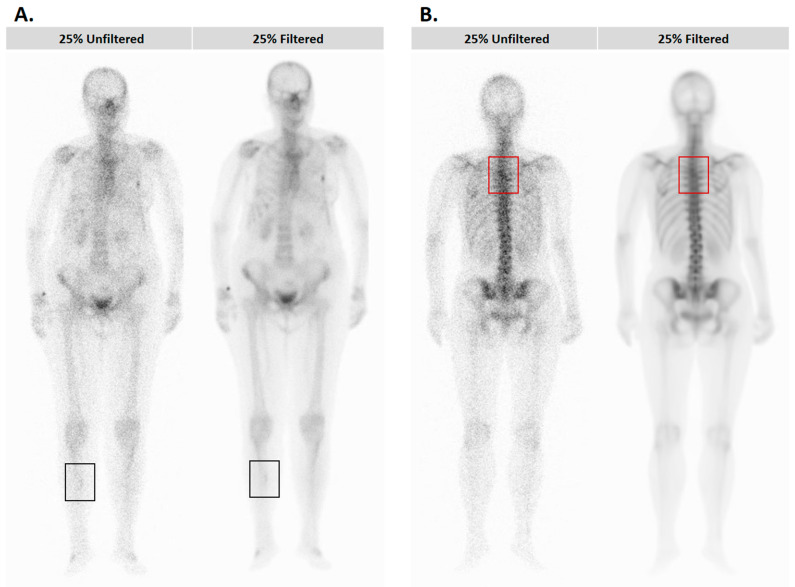
Patient images in which the application of the BS-AI filter resulted in a highlighted (**A, black rectangle**) or a disappearing (**B, red rectangle**) lesion. There was no consensus among the three physicians in either case, so the presence/absence of these lesions is debatable.

**Figure 4 diagnostics-14-02686-f004:**
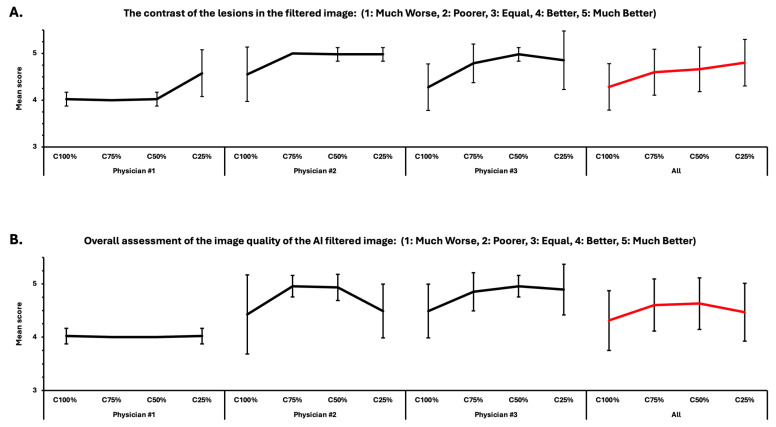
Visual assessment of three different nuclear medicine physicians is shown. Assessment of lesion contrasts (**A**) as well as overall image quality (**B**) of the filtered images compared to the unfiltered images are shown. Mean scores ± SD for different levels of degradation given individually (**black graphs**) and combined (**red graph**) are present.

**Figure 5 diagnostics-14-02686-f005:**
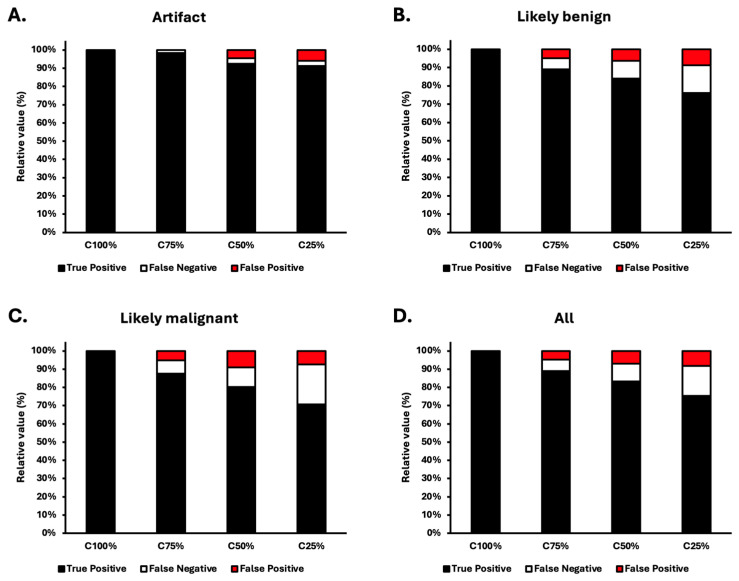
Lesion detection by the BS-AI annotator after filtering the total-count (C100%) and lower-count (C75%, C50%, and C25%) images. All lesions found by the detector in the filtered total-count images were considered as reference (TP), extra lesions compared to these were defined as FP, and lesions missed from these lesions were defined as FN. Distribution of detecting lesions as artifacts (**A**), likely benign (**B**), or likely malignant (**C**), shown separately and all together (**D**). Percentages of each category are shown.

**Figure 6 diagnostics-14-02686-f006:**
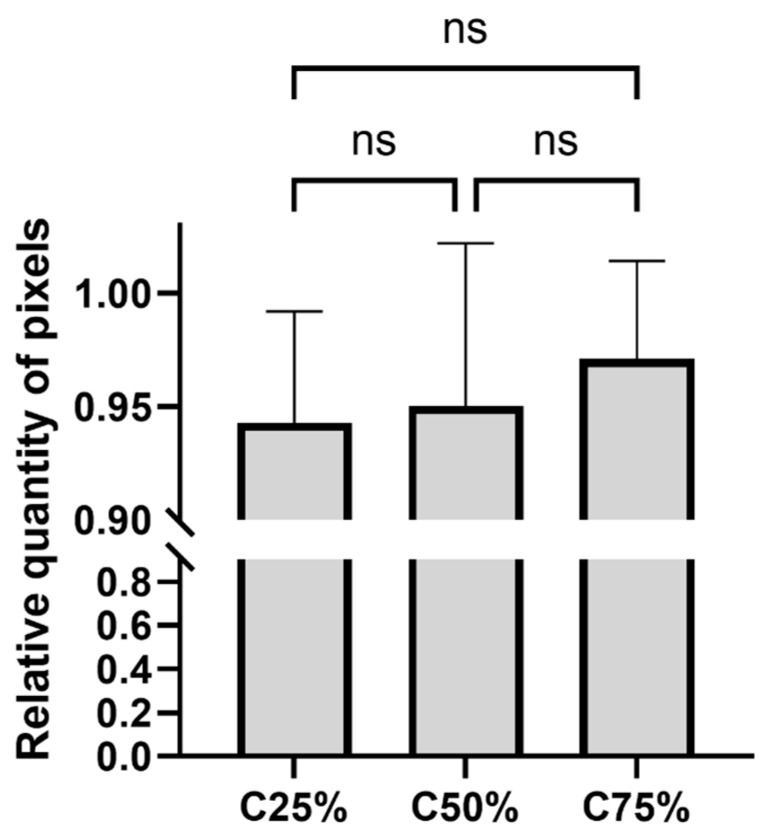
Mean pixel counts relative to the number of pixels marked in the C100% images, ±SD. Bars marked with “ns” indicate no significant difference (*p* > 0.05) in the pixel numbers between the filtered images of different counts. (Tukey test; ns: not significant, *p* > 0.05).

## Data Availability

The dataset used and analyzed in the current study is available from the corresponding author on reasonable request.

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
