# Peer review of "AI-Based Noise-Reduction Filter for Whole-Body Planar Bone Scintigraphy Reliably Improves Low-Count Images"

_diagnostics, 2024, doi:10.3390/diagnostics14232686_

Round 1
Reviewer 1 Report
Comments and Suggestions for Authors
In the mansucript "AI-Based Noise Reduction Filter for Whole-Body Planar Bone 2 Scintigraphy Reliably Improves Low-Count Images" a new AI-based algorithm to improve bone scan images with low count statistics is presented and validated. An appropriate number of 1215 data sets was used for training.
The paper is well written and good to understand, however I have just two issues that should be addressed:
- Fig. 2: details are diffucult so see, perhaps include a zoom of an interesting region (pelvic area) so see more details.
- in the discussion section the potential transfer of this method to SPECT images should be discussed more in detail.
Reviewer 2 Report
Comments and Suggestions for Authors
General Comments:
AI applications in denoising are commonly used in imaging, including nuclear medicine imaging. This paper is in principle interesting, however many important questions remain unanswered and the methodology is not always described in an understandable manner. However, I am positive that the authors can substantially improve the manuscript in both readability and content.
It becomes clear that the writing author is not a native speaker, many incorrect or at least inadequate English expressions are used; therefore, language editing is strongly advised.
An inappropriate Likert scale is used, this seems due to a language problem, however (see below, comment to L105/107)
The abstract does not stand alone (what it should), many things from the abstract only become clear after reading the full manuscript. This needs to be corrected.
There may be a bias in the evaluation by human readers. The reading procedure is not described, see comment below. This needs to be described in much more detail, and explained thoroughly.
The number of patient image pairs does used (47) not correspond to the selection scheme described: 3(BMI)*3(age)*2(gender)*3(#lesions)=54
Specific comments:
P1:
L17: “on 47 99mTc-MDP bone scintigraphy images which”: 47 images or patients? Neither nor, I guess it is 47 image pairs
L22: “The total number of lesions detected by the BS-AI annotator in the BS-AI filtered low-count images were compared to the original filtered images”: only the number or also if they are at the same location, i.e., same suspected lesions?
L23: “The total number of pixels labeled as lesions in the filtered low-count images relative to the number of pixels in the original filtered images were compared;“: so both, total number and total number of pixels were compared? This is not clear from reading the abstract alone (it is however clear from reding the whole paper, but abstract must be clear without then reading the whole paper)
L27: “The similarities of lesions detected by BS-AI annotator compared to filtered total-count images were 89%, 83%, 75%”: what is "similiarities"? Numbers and sites? Pixels? What exactly does this mean?
L29: “There was no significant difference in the number of annotated pixels between filtered images with different counts;”: what does it mean that the number of pixels is same if they are in different locations? So I guess this is not what authors want to state - it is clear from reding tho paper, but not from the abstract alone. Also, „significant“ requires a p value reported
L33: „possible to reduce the injected dose or acquisition time”: how much? Provide numbers, how much reduction counts can be compensated?
L43: “nuclear radiologists”: nuclear medicine physician
P2:
L90: “47 pairs of anterior and posterior view: 47 image pairs? Shouldn't it be 3(BMI)*3(age)*2(gender)*3(#lesions)=54?
L91: “with one image selected at random for each category”: describe how this was done
P3:
L103ff: “Is there any lesion which is missed from the AI filtered image? (1: No, 2: Undecided, 3: Yes);”: So if the lesions detected on the 100%counts image is taken as gold standard (ground truth), why would one not be able to decie in a binary manner (yes/no)?
L105: “The contrast of the lesions in the filtered image: (1: Worse, 2: Poor, 3: Equal, 4: Good, 5: Much Better”: Why is it appropriate to use "Equal" "Good" as different ratings? "Good" is meant as "slightly better than"? Good is not an appropriate specification in a Likert scale comparing images. Worse, poor is actually also not, but ok since will be interpreted as worse than, and poorer than (what would be more appropriate)
L107: same
P4:
L159: “During visual assessment the nuclear medicine physicians did not find any vanishing or new lesions after filtering the original images and the images degraded to 75% and 50% of the original counts.” The reading scenario is not desribed. How is it assured that readers did not recognice a patient image and thus be biased because they have seen the higher count image before? This is a critical issue!
P5:
L162: “Similarly, two physicians each detected one image that contained a new lesion after filtering, which was not visible in the original image.”: This is important! Elaborate on that finding! Details please!
L169, Figure 3: “Assessment of lesion contrasts” and “of the filtered images compared to the non-filtered images are shown.”: compared to what? Are the 75,50,25% count rate filtered images compared to the respective unfiltered images or to the 100% count rate unfilted image which is the "ground truth"? In case thea are compared to the unfiltered images with the same count rate, a comparison to the 100% count rate unfiltered image is missing to estimate the potential count rate/activity reduction potential
L174: „Assessment of similarities in lesion detection by the BS-AI annotator indicated that artifacts were the lesions most consistently assessed compared to the filtered total“: don't understand this sentence. Pls rephrase. "artifacts were the lesions" does not make sense to me. lso I do not understand what exactly "similiarities" means, please define
P6:
L183, Figure 4: In figure 4 we only see results from filtered images. Add 100% count rate unfiltered. This is important to get an idea of the performance of the filtering procedure
L190: “Similarities to C100% images“: again, "similiarity" is somehow defined to be a quantitative number. Still not understanding what exactly this similiarity as a number means and how exactly it is defined. This needs to be explained and define in the Materials section
Comments on the Quality of English Language
needs to be improved, some instances stated in the report (but not all)
Round 2
Reviewer 2 Report
Comments and Suggestions for Authors
From the point by point reply to my queries I conclude that they have been answered satisfactorily, and thus endorse publication